# Intercalated disc protein Xinβ is required for Hippo-YAP signaling in the heart

Haipeng Guo[1,2], Yao Wei Lu [1], Zhiqiang Lin[1,3], Zhan-Peng Huang[1,4], Jianming Liu[1], Yi Wang [1], Hee Young Seok[1,5], Xiaoyun Hu[1], Qing Ma[1], Kathryn Li[1], Jan Kyselovic[6], Qingchuan Wang[7,8], Jenny L.-C. Lin[7], Jim J.-C. Lin[7], Douglas B. Cowan [1], Francisco Naya[9], Yuguo Chen[2], William T. Pu[1,10] & Da-Zhi Wang [1,10✉]

Intercalated discs (ICD), specific cell-to-cell contacts that connect adjacent cardiomyocytes, ensure mechanical and electrochemical coupling during contraction of the heart. Mutations in genes encoding ICD components are linked to cardiovascular diseases. Here, we show that loss of Xinβ, a newly-identified component of ICDs, results in cardiomyocyte proliferation defects and cardiomyopathy. We uncovered a role for Xinβ in signaling via the Hippo-YAP pathway by recruiting NF2 to the ICD to modulate cardiac function. In *Xinβ* mutant hearts levels of phosphorylated NF2 are substantially reduced, suggesting an impairment of Hippo-YAP signaling. Cardiac-specific overexpression of YAP rescues cardiac defects in *Xinβ* knock-out mice—indicating a functional and genetic interaction between Xinβ and YAP. Our study reveals a molecular mechanism by which cardiac-expressed intercalated disc protein Xinβ modulates Hippo-YAP signaling to control heart development and cardiac function in a tissue specific manner. Consequently, this pathway may represent a therapeutic target for the treatment of cardiovascular diseases.

[1] Department of Cardiology, Boston Children's Hospital, Harvard Medical School, 320 Longwood Avenue, Boston, MA 02115, USA. [2] Department of Critical Care and Emergency Medicine, Key Laboratory of Cardiovascular Remodeling and Function Research, Chinese Ministry of Education and Chinese Ministry of Health, Qilu Hospital, Cheeloo College of Medicine, Shandong University, Jinan 250012, China. [3] Masonic Medical Research Institute, 2150 Bleecker St, Utica, NY 13501, USA. [4] Department of Cardiology, Center for Translational Medicine, The First Affiliated Hospital, NHC Key Laboratory of Assisted Circulation, Sun Yat-sen University, Guangzhou, China. [5] Institute of Life Sciences and Biotechnology, Korea University, Seoul, Korea. [6] Department of Internal Medicine, Faculty of Medicine, Comenius University, Ruzinovska 6, 826 06 Bratislava, Slovak Republic. [7] Department of Biology, University of Iowa, Iowa City, IA 52242, USA. [8] Department of Medicine, Johns Hopkins University School of Medicine, Baltimore, MD 20215, USA. [9] Department of Biology, Boston University, Boston, MA 02215, USA. [10] Harvard Stem Cell Institute, Harvard University, Cambridge, MA 02138, USA. ✉email: Da-Zhi.Wang@childrens.harvard.edu

Cardiovascular diseases remain the leading cause of death in humans, and the molecular mechanisms underlying these devastating conditions remain largely elusive. The intercalated disc (ICD), which is a subcellular structure that connects neighboring cardiomyocytes, is essential for the functional integrity of the cardiac syncytium—mechanically tethering cardiomyocytes to one another and facilitating propagation of electrical signals that coordinate contraction of the heart[1]. Interestingly, while many prior reports demonstrate the importance of ICDs to the organization of the myocardium, relatively little is known about how these cell-to-cell junctions transmit information between cardiac muscle cells to modulate gene expression and cardiac function[2].

The Xin-repeat containing adapter proteins Xinα and Xinβ, also called XIRP1 and XIRP2, are encoded by paralogous genes and bind various adherens junction proteins including actin, N-cadherin, α-actinin, and β-catenin[3–7]. As a result, Xinα and Xinβ are primarily located in the ICD of adult cardiomyocytes[8]. These proteins also play an important role during early cardiac development[4,9] and may be pivotal in the pathogenesis of some forms of heart disease[6]. For instance, the human homolog of mouse $Xin\beta$ (h$Xin\beta$) was mapped to a locus associated with cardiomyopathy[10]. Mutations in $Xin\alpha$ and $Xin\beta$ have also been found in patients with cardiomyopathy, underscoring their potential to contribute to the initiation and progression of cardiovascular diseases[9,10].

The Hippo-YAP is a highly conserved cellular regulatory pathway that is important for development and disease[11]. Genetic and biochemical studies have established that cellular growth and differentiation signals are transmitted through a series of protein kinase cascades to modulate downstream gene expression[12]. Dysregulation of the expression of this important pathway and/or genetic mutations of components of this pathway are linked to human diseases, including cancer[13,14]. Studies have found that the Hippo-YAP pathway plays an important role in cardiomyocyte proliferation and heart development[14–19]. In this study, we reveal the unexpected finding that function of the ICD protein Xinβ is mediated by the Hippo-YAP pathway to control heart development and cardiac function.

## Results

**Loss of $Xin\beta$ alters Hippo-YAP pathway gene expression in the heart.** Consistent with prior studies[4], we found that germline deletion of $Xin\beta$ ($Xin\beta^{KO}$) in mice resulted in postnatal lethality. Mutant mice die before weaning and display both developmental and cardiac structural defects. To define the molecular mechanisms underlying the observed cardiac defects in $Xin\beta^{KO}$ mice, we performed RNA-seq on ventricular tissue from postnatal day 7.5 (P7.5) $Xin\beta^{KO}$ and control hearts. A smear plot shows the distribution of gene expression between $Xin\beta^{KO}$ and control hearts, whereas $Xin\beta$ (also called $Xirp2$) was the most significantly downregulated gene (Fig. 1a). Gene expression analyses showed that 717 genes were differentially expressed ($p < 0.01$), with 373 genes increased and 344 genes reduced in $Xin\beta^{KO}$ mouse hearts (Fig. 1b). Gene set enrichment analysis (GSEA) identified gene ontology (GO) terms that were enriched for upregulated or downregulated genes. Among the GO terms for genes downregulated in $Xin\beta^{KO}$ mouse hearts were those related to chromosome segregation, chromosome condensation, DNA replication, and cell division (Fig. 1c). In contrast, terms related to response to type I interferon, and positive regulation of inflammatory response, were significantly enriched for genes upregulated in $Xin\beta^{KO}$ hearts (Fig. 1c), indicating an increase of inflammatory response in $Xin\beta$ mutant hearts. We verified the dysregulated expression of selected genes related to cell

proliferation and inflammation by qRT-PCR (Fig. 1d, e). Interestingly, our GSEA analysis revealed that the "YAP-conserved signature" was substantially downregulated in P7.5 $Xin\beta^{KO}$ hearts (Fig. 1f). Using qRT-PCR assays, we confirmed downregulation of YAP pathway genes in $Xin\beta^{KO}$ hearts (Fig. 1g).

**Inverted gene expression regulated by $Xin\beta$ mutation and YAP overexpression.** The finding that the YAP pathway was dysregulated in $Xin\beta^{KO}$ hearts raised the possibility that Xinβ controls cardiac gene expression and cardiac function through regulation of the Hippo-YAP pathway. We reasoned that the top upregulated and downregulated genes in $Xin\beta^{KO}$ hearts would also be affected in YAP gain- or loss-of-function cardiomyocytes. To this end, we compared the transcriptome signatures generated from the $Xin\beta^{KO}$ hearts with those generated from YAP-overexpressing cardiomyocytes[17] (Fig. 1h). Indeed, scatter plot analysis revealed differential gene expression in $Xin\beta^{KO}$ hearts vs. YAP-overexpressing cardiomyocytes (Fig. 1i). This observation is further supported by GSEA analysis, which showed that similar functional terms were enriched in $Xin\beta^{KO}$ upregulated and downregulated genes because of $YAP$ overexpression, and vice versa (Fig. 1i). In particular, we found that genes related to "E2F targets, G2M checkpoint, and mitotic spindle", the top downregulated GO term in $Xin\beta^{KO}$ hearts, was among the significant GO terms for upregulated genes in $YAP$-overexpressing cardiomyocytes (Fig. 1i and Supplementary Fig. 1a). $YAP$-induced expression of cell-cycle-related genes was confirmed by qRT-PCR (Supplementary Fig. 1b). Conversely, the GO term "interferon-alpha response, interferon-gamma response," among the most enriched for upregulated genes in $Xin\beta^{KO}$ hearts, was among the most strongly enriched for downregulated genes in $YAP$-overexpressing cardiomyocytes (Fig. 1i and Supplementary Fig. 1c). We also confirmed $YAP$-induced downregulation of these innate immune response genes by qRT-PCR (Supplementary Fig. 1d). These results indicate that Xinβ is involved in the regulation of cardiomyocyte proliferation and inflammatory responses, likely by modulating the expression and function of the YAP signaling pathway.

**Decreased cardiomyocyte proliferation in $Xin\beta$ mutant mice.** The above results suggest that Xinβ regulates cardiomyocyte proliferation in developing hearts. Indeed, we found reduced cardiomyocyte proliferation/nuclear division in $Xin\beta^{KO}$ hearts, as demonstrated by decreased levels of EdU (5-ethynyl-2′-deoxyuridine) incorporation (Fig. 2a) and phosphorylated histone H3 (pH3) (Fig. 2b), consistent with our prior report[4]. $Xin\beta^{KO}$ hearts were substantially smaller at P7.5 when compared with littermate controls (Fig. 2c). There was an overall reduction of ventricle wall thickness and the size of the ventricular chamber, without any obvious fibrosis (Fig. 2d). In addition to the proliferation defect, we observed an increase in apoptosis in postnatal $Xin\beta^{KO}$ hearts (Supplementary Fig. 2a, b), and increased expression of genes related to apoptosis (Supplementary Fig. 2c). In contrast, Ad-$YAP$ overexpression reduced apoptotic gene expression in cardiomyocytes (Supplementary Fig. 2d).

Global $Xin\beta^{KO}$ mutant mice die postnatally due to cardiac defects; therefore, we asked whether loss of Xinβ in cardiomyocytes at a later stage would lead to a similar phenotype. We used our recently reported adeno-associated virus serotype 9 (AAV9)-mediated CRISPR/Cas9 somatic mutagenesis platform[20,21] to inactivate $Xin\beta$ in postnatal cardiomyocytes. Our previous study demonstrated that high-dose AAV9-mediated ablation of an essential gene resulted in lethal heart failure[20]. We injected neonatal Rosa$^{Cas9GFP/Cas9GFP}$ mice with AAV9 carrying a $cTNT$-promoter-driven $Cre$, which activates Cas9 expression in

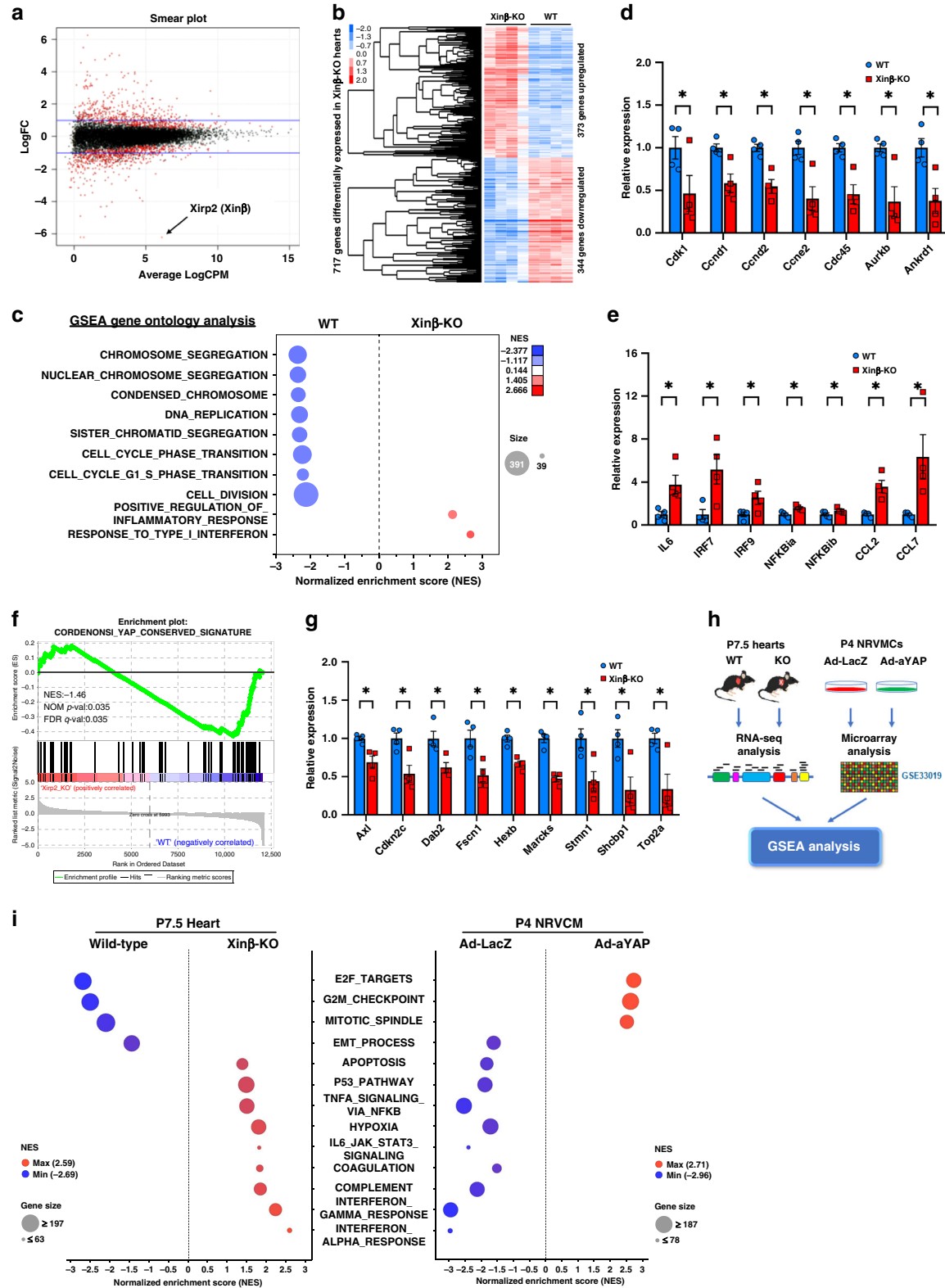

cardiomyocytes from the *Rosa* locus, together with two U6 promoter-driven guide RNAs (gRNAs), designed to specifically target mouse *Xinβ* (referred to as AAV-*Xinβ*^sgRNA). As a control, we injected AAV-*cTNT-Cre* without gRNA (Fig. 2e and Supplementary Fig. 3a). A neonatal administration of AAV-*cTNT-Cre* or AAV-*Xinβ*^sgRNA transduced 80% of cardiomyocytes as determined by GFP expression in the heart (Supplementary Fig. 3b). We confirmed >80% reduction of *Xinβ* mRNA

expression in mouse hearts after AAV-*Xinβ*^sgRNA injection (Supplementary Fig. 3c). Immunostaining indicated that Xinβ was not detected in GFP-positive cardiomyocytes (Supplementary Fig. 3d). Unlike global *Xinβ*^KO mice, which all died prior to weaning, AAV-*Xinβ*^sgRNA-injected mice lived to adulthood (Supplementary Fig. 3e). Echocardiography showed that AAV-*Xinβ*^sgRNA-treated mice had modestly reduced cardiac function at 1–6 months of age (Fig. 2f and Supplementary Fig. 3f). We

**Fig. 1 Loss of *Xinβ* results in alteration of the Hippo-YAP signaling pathway in the heart. a** Smear plot showing the log fold change and average abundance of each gene. Differentially expressed genes are marked red; the expression of *Xinβ* (also called *Xirp2*) is among the mostly downregulated in P7.5 (day 7.5) *Xinβ*$^{KO}$ hearts; **b** hierarchical clustering heatmap of differentially expressed genes; **c** gene set enrichment analysis (GSEA) with gene ontology gene-sets reveals molecular pathways related to cell proliferation and inflammation dysregulated in P7.5 *Xinβ*$^{KO}$ hearts; **d** qRT-PCR of cell-cycle-related genes in P7.5 *Xinβ*$^{KO}$ and control hearts. $N = 4$ biologically independent samples, *$P < 0.05$. **e** qRT-PCR of genes related to immune and inflammatory response in P7.5 *Xinβ*$^{KO}$ and control hearts. $N = 4$ biologically independent samples, *$P < 0.05$. **f** Enrichment plot showing downregulation of the YAP-conserved pathway; **g** qRT-PCR of YAP pathway genes in P7.5 *Xinβ*$^{KO}$ and control hearts. $N = 4$ biologically independent samples, *$P < 0.05$. **h** Workflow of comparative analysis of dysregulated genes between *Xinβ*$^{KO}$ hearts and Ad-*YAP* cardiomyocytes; **i** GSEA with MSigDB Hallmark gene sets (v 6.2) displaying differentially regulated pathways between *Xinβ*$^{KO}$ hearts and Ad-a*YAP* cardiomyocytes.

isolated cardiomyocytes from AAV-*Xinβ*$^{sgRNA}$-injected hearts and found cardiomyocytes without Xinβ immunoreactivity appeared to have normal morphology and sarcomere organization (Fig. 2g). Together, these results indicate that loss of Xinβ does not affect the morphology of adult cardiomyocytes, while it is required for neonatal cardiomyocyte proliferation.

**Cardiac *YAP*$^{S127A}$ overexpression rescues *Xinβ*$^{KO}$ defects.** We hypothesized that YAP mediates the function of Xinβ in the heart to regulate cardiomyocyte proliferation and cardiac function. We used AAV9 to overexpress Flag-tagged *YAP* (3XFLAG-*YAP* [S127A]—referred to as AAV-*YAP*$^{S127A}$ (or a*YAP* in the figures) in which the inhibitory Hippo phosphorylation site serine 127 is mutated to alanine[19]) in cardiomyocytes. Consistent with our previous report[19], AAV-*YAP*$^{S127A}$ was predominantly localized in the nuclei of cardiomyocytes (Supplementary Fig. 4a). We administered AAV-*YAP*$^{S127A}$ or AAV-*GFP* control to *Xinβ*$^{KO}$ or control mice at P1 (Fig. 3a). Remarkably, almost 40% of the AAV-*YAP*$^{S127A}$-treated *Xinβ*$^{KO}$ mice (hereafter referred to as *Xinβ*$^{KO}$/*YAP*$^{S127A}$ mice) survived to adulthood; in sharp contrast, all control AAV-*GFP*-injected *Xinβ*$^{KO}$ mice died before 20 days of age (Fig. 3b). The gross morphology of the rescued *Xinβ*$^{KO}$/*YAP*$^{S127A}$ mice was comparable to that of control mice (Fig. 3c). Furthermore, the morphology, histology, and function of *Xinβ*$^{KO}$/*YAP*$^{S127A}$ hearts were indistinguishable from their littermate controls (Fig. 3d, e and Supplementary Fig. 4b). As we described previously[18,19,22], *YAP*$^{S127A}$ overexpression was well tolerated in wild-type (WT) mice and did not significantly alter their cardiac function (Supplementary Fig. 4b).

To better understand the molecular mechanisms by which activated YAP rescued *Xinβ*$^{KO}$ defects in the heart, we performed RNA-seq on postnatal day 7.5 (P7.5) *Xinβ*$^{KO}$/*YAP*$^{S127A}$ and control (WT/*GFP*; WT/*YAP*$^{S127A}$; *Xinβ*$^{KO}$/*GFP*) hearts. Principal component analysis (PCA) showed that the transcriptome of *Xinβ*$^{KO}$ hearts clearly segregated from the other groups (Fig. 3f). Overexpression of *YAP*$^{S127A}$ normalized the gene expression of *Xinβ*$^{KO}$/*YAP*$^{S127A}$ so that it overlapped with the other control groups (Fig. 3f). The normalization of gene expression was further demonstrated by hierarchical clustering of differentially expressed genes, which showed that genes upregulated and downregulated in *Xinβ*$^{KO}$ hearts became indistinguishable from controls in *Xinβ*$^{KO}$/*YAP*$^{S127A}$ hearts (Fig. 3g). GSEA revealed that the functional categories of E2F targets, G2M checkpoint, and mitotic spindle were completely restored in *Xinβ*$^{KO}$/*YAP*$^{S127A}$ hearts, while genes related to Bile acid metabolism, fatty acid metabolism, adipogenesis, and others found to be upregulated in *Xinβ*$^{KO}$ hearts were repressed in Xinβ$^{KO}$/*YAP*$^{S127A}$ hearts (Fig. 3h). While the "YAP-conserved signature" was highly enriched for genes downregulated in *Xinβ*$^{KO}$/*GFP* compared to WT/*GFP*, it was strongly enriched for genes upregulated in *Xinβ*$^{KO}$/*YAP*$^{S127A}$ compared to *Xinβ*$^{KO}$/*GFP* (Fig. 3i). Specifically, expression of cell proliferation and apoptosis genes was corrected in *Xinβ*$^{KO}$/*YAP*$^{S127A}$ hearts (Fig. 3j and Supplementary Fig. 4c, d). Consistent with the expression profile, we found that *YAP*$^{S127A}$ overexpression was sufficient to attenuate cardiomyocyte proliferation defects observed in

*Xinβ*$^{KO}$ hearts (Fig. 3k and Supplementary Fig. 4e, f). Together, our data show that an active form of YAP (i.e., *YAP*$^{S127}$) is sufficient to rescue defects found in *Xinβ*$^{KO}$ mice, supporting the conclusion that YAP functions downstream of Xinβ to regulate cardiac gene expression and function.

To further confirm the function of Xinβ in cardiomyocyte proliferation and the involvement of YAP in this process, we knocked down Xinβ in primary neonatal rat ventricular cardiomyocytes (NRVMs). As expected, inhibition of Xinβ reduced cardiomyocyte proliferation, marked by the reduction in EdU incorporation and pH3 staining (Supplementary Fig. 5a, b). Adenovirus-mediated overexpression of active YAP (Ad-*YAP*$^{S127A}$) enhanced WT cardiomyocyte proliferation (Supplementary Fig. 5a, b) and completely rescued the proliferation defect caused by Xinβ knockdown (Supplementary Fig. 5). The expression of cell-cycle regulatory genes was lower in Xinβ knockdown cardiomyocytes and normalized in rescued cardiomyocytes (Supplementary Fig. 5c). These data, together with the above in vivo studies, suggest that YAP functions downstream of Xinβ to regulate cardiomyocyte proliferation.

**Xinβ interacts with YAP pathway component NF2.** Next, we asked how Xinβ regulates the Hippo-YAP pathway. In particular, we tested whether loss of Xinβ protein alters the expression level and/or subcellular location of the Hippo-YAP signaling proteins in the heart. Among the components of the Hippo-YAP pathway tested, we found that the expression of phosphorylated neurofibromin 2 (P-NF2) was reduced in P7.5 *Xinβ*$^{KO}$ hearts, whereas total NF2 (T-NF2) protein was not altered (Fig. 4a, b). In contrast, phosphorylated YAP (P-YAP) protein was increased and total YAP was unaffected in *Xinβ*$^{KO}$ hearts (Fig. 4a, b). Decreased P-NF2 protein was confirmed by lower P-NF2 immunoreactivity in P7.5 *Xinβ*$^{KO}$ hearts compared with controls (Fig. 4c). It has been well documented that activation of the Hippo pathway kinase cascade results in phosphorylation of YAP protein, leading to its degradation and reduction in nuclear levels[11,13]. We examined the cellular location of YAP proteins and found that loss of Xinβ decreased YAP nuclear localization in cardiomyocytes (Fig. 4d). These results are consistent with the findings that loss of Xinβ results in decreased YAP target gene expression (Fig. 1f, g) and cardiomyocyte proliferation (Fig. 2a, b).

NF2 protein has been reported to be located in the cell membrane, cytoplasm, and nucleus of cardiomyocytes[23]. We examined the subcellular location of NF2 in cardiomyocytes and found it was predominantly located on the cell membrane. At ICDs of adult cardiomyocytes, it co-localized with Xinβ (Fig. 4e). We then asked whether NF2 and Xinβ proteins interact. Co-immunoprecipitation (Co-IP) assays with epitope-tagged proteins showed Xinβ forms a complex with NF2, and that this interaction requires a domain of Xinβ between amino acid residues 384 and 680 (Fig. 4f). We confirmed this interaction by co-immunoprecipitating either epitope-tagged Xinβ or NF2 and using an independent, overlapping Xinβ fragment (Fig. 4g–i) as well as showing endogenous Xinβ interacted with NF2 in WT, but not *Xinβ*$^{KO}$ hearts (Fig. 4j).

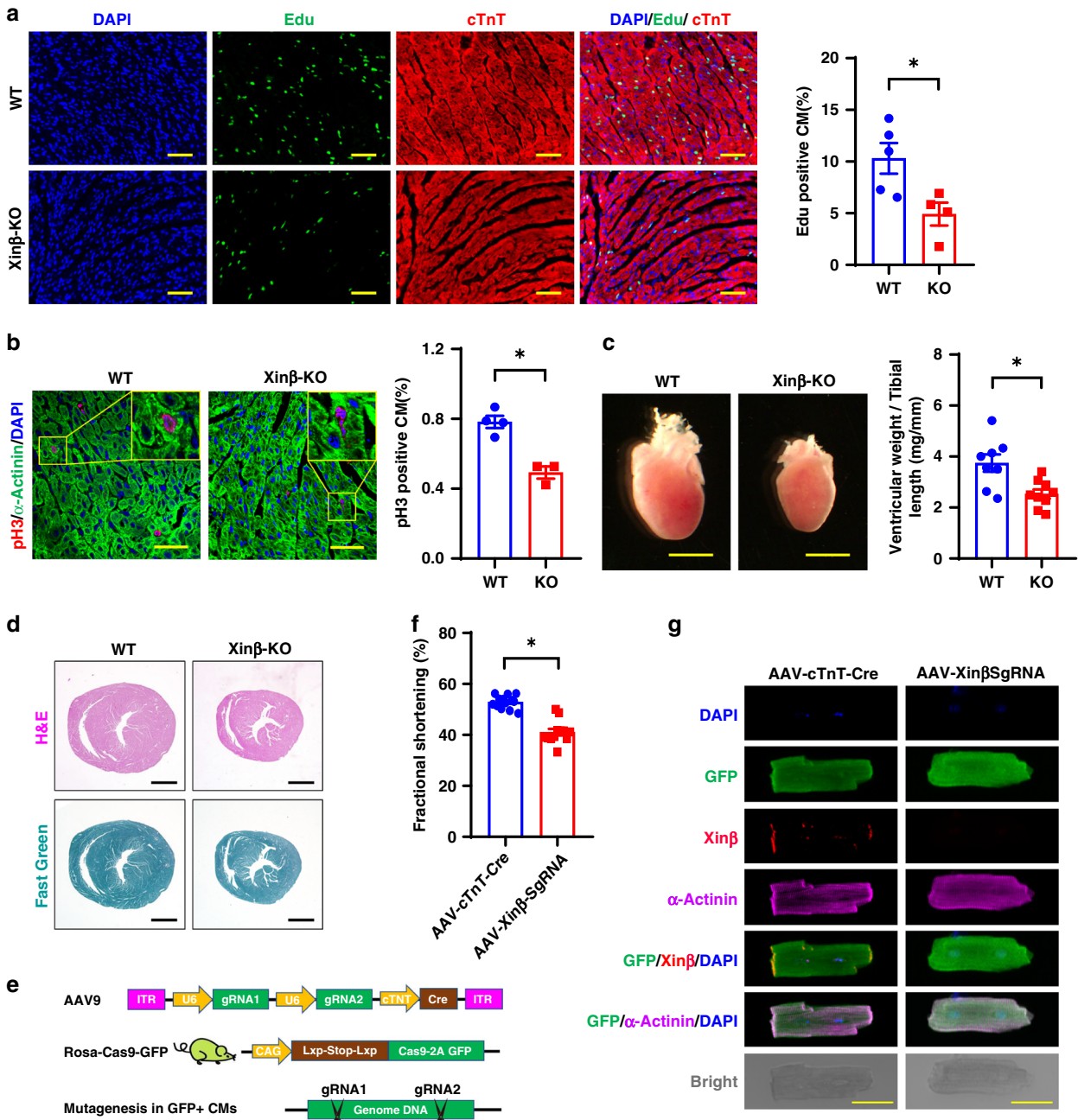

**Fig. 2 Decreased cardiomyocyte proliferation in *Xinβ*[KO] hearts. a** Edu incorporation in P7.5 *Xinβ*[KO] and control hearts. DAPI marks nuclei, cTnT labels cardiomyocytes. Quantification in right panel. Scale bars = 100 μm. N = 4 and 5 biologically independent samples, respectively, *P < 0.05. **b** pH3 staining of P7.5 *Xinβ*[KO] and control hearts. DAPI marks nuclei, α-actinin labels cardiomyocytes. Quantification in right panel. Scale bars = 40 μm. N = 4 and 3 biologically independent samples, respectively, *P < 0.05. **c** Heart morphology (left panel) and ventricle weight vs. tibial length ratio (right panel) of P7.5 *Xinβ*[KO] and control hearts. Scale bars = 2 mm. N = 9 and 8 biologically independent samples, respectively,*P < 0.05. **d** Hematoxylin and eosin (red) and Fast Green (green) staining of p7.5 *Xinβ*[KO] and control hearts. Scale bars = 1 mm. **e** AAV9-based CRISPR/Cas9 strategy to mutate the *Xinβ* gene in postnatal mouse hearts; **f** percent fractional shortening (FS) change in 4-week-old AAV-*cTnT-Cre* and AAV-*Xinβ-sgRNA*-treated mice. N = 13 and 12 biologically independent samples, respectively, *P < 0.05. **g** Morphology and immunostaining of isolated cardiomyocytes from hearts of adult *Xinβ*[KO] and control mice. Scale bars = 50 μm.

Given that loss of *Xinβ* results in severe cardiac defects in mice, we asked whether the expression of human *Xinβ* (h*Xinβ*) is altered in cardiovascular disease. In the hearts of dilated cardiomyopathy (DCM) patients, h*Xinβ* transcript was reduced (Fig. 4k). Consistent with this observation, expression of cardiomyopathy marker genes *NPPA* and *NPPB* were dramatically induced in the hearts with DCM (Fig. 4k). Together, these results indicate that dysregulation of h*Xinβ* expression is associated with human cardiac disease.

## Discussion

The ICD is a unique structure that is important for cardiomyocyte communication, heart development, and cardiac function[1]. Genetic studies have revealed that mutations in genes encoding many of the ICD proteins are associated with human heart diseases[25,26]. In addition, numerous studies have shed light on the development and maturation of the ICD. Of particular relevance to the present study, the developmental expression and intracellular location of Xinα and Xinβ in neonatal and adult hearts has

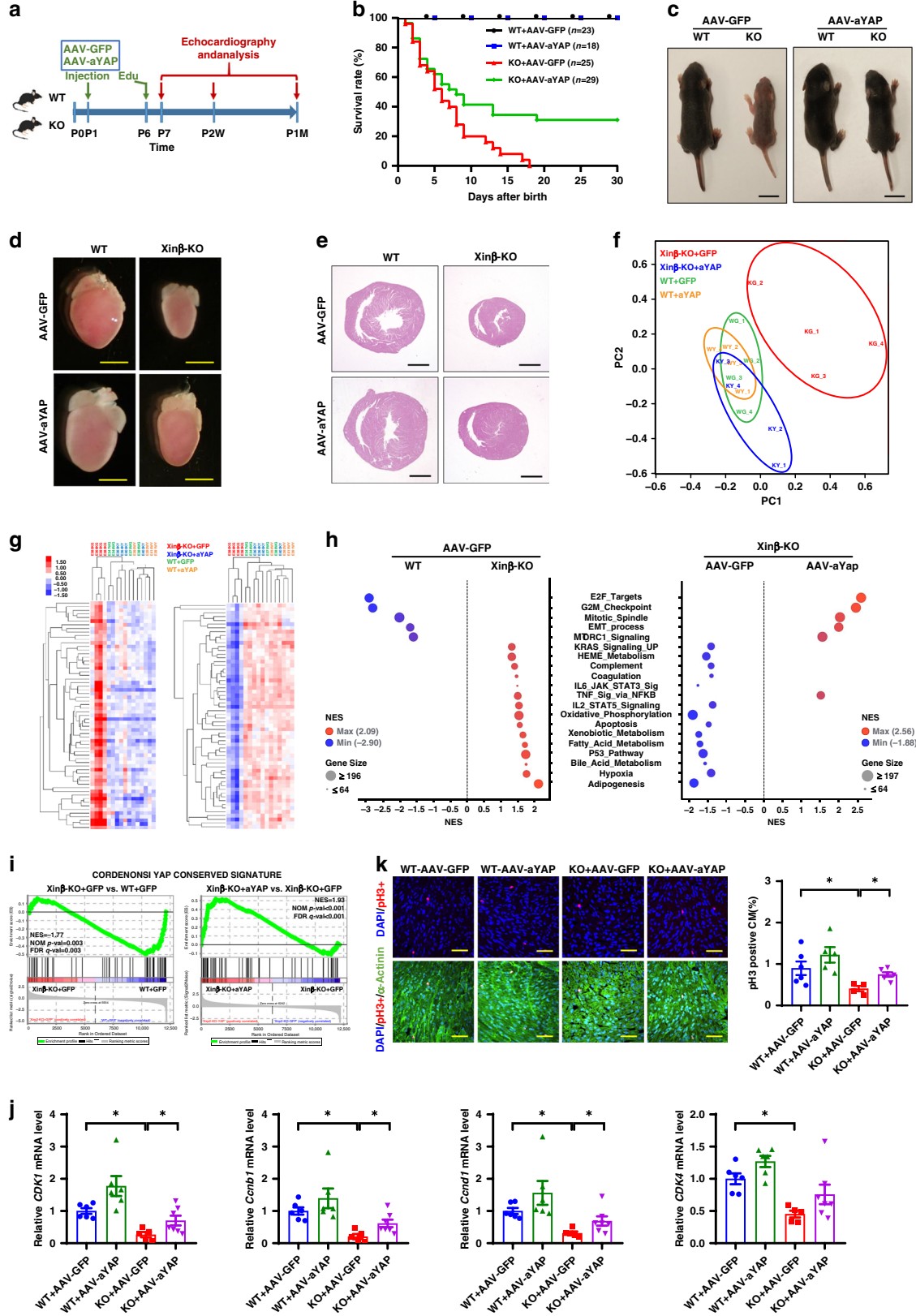

been described[6,24]. These studies showed maturation of the ICD was prevented in neonatal cardiomyocytes from Xinβ-deficient mouse hearts, as evidenced by the failure of the adherens junction protein N-cadherin to reorganize from the lateral surface of cardiomyocytes to their terminus.

Our current loss of function and rescue studies establish that the ICD-associated protein Xinβ is required for postnatal cardiomyocyte proliferation, which is likely mediated by the Hippo-YAP pathway in the heart. We observed that a substantial portion of $Xin\beta^{KO}$ mice was rescued to adulthood by neonatal delivery of

**Fig. 3 Heart-specific *YAP*<sup>S127A</sup> overexpression rescues cardiac defects in *Xinβ*<sup>KO</sup> mice. a** Diagram of the workflow. **b** Survival curve of WT and *Xinβ*<sup>KO</sup> mice injected with AAV-*aYAP* (*YAP*<sup>S127A</sup>) or control AAV-*GFP*. **c** Gross morphology of P7.5 WT and *Xinβ*<sup>KO</sup> mice injected with AAV-*aYAP* or control AAV-*GFP*. Scale bars = 1 cm. **d** Heart gross morphology and, **e** histology, of P7.5 WT and *Xinβ*<sup>KO</sup> mice injected with AAV-*aYAP* or control AAV-*GFP*. Scale bars = 2 mm (**d**) and 1 mm (**e, f**). Principal component analysis of all expressed genes in P7.5 WT and *Xinβ*<sup>KO</sup> mice injected with AAV-*aYAP* or control AAV-*GFP*. **g** Hierarchical clustering heatmap of differentially expressed genes. **h** Gene set enrichment analysis (GSEA) of differentially regulated pathways P7.5 WT and *Xinβ*<sup>KO</sup> mice injected with AAV-*YAP* or control AAV-*GFP*. **i** Enrichment plot of rescued YAP-conserved signatures in *Xinβ*<sup>KO</sup> hearts injected with AAV-*aYAP*. **j** qRT-PCR of cell-cycle-related genes in the hearts of WT and *Xinβ*<sup>KO</sup> mice injected with AAV-*aYAP* or control AAV-*GFP*. N = 6 or 7 biologically independent samples, *P < 0.05. **k** pH3 staining of hearts from P7.5 *Xinβ*<sup>KO</sup> and control mice injected with AAV-*aYAP* or control AAV-*GFP*. DAPI marks nuclei, α-actinin labels cardiomyocytes. Quantification in right panel. Scale bars = 40 μm. N = 5 or 6 biologically independent samples, *P < 0.05.

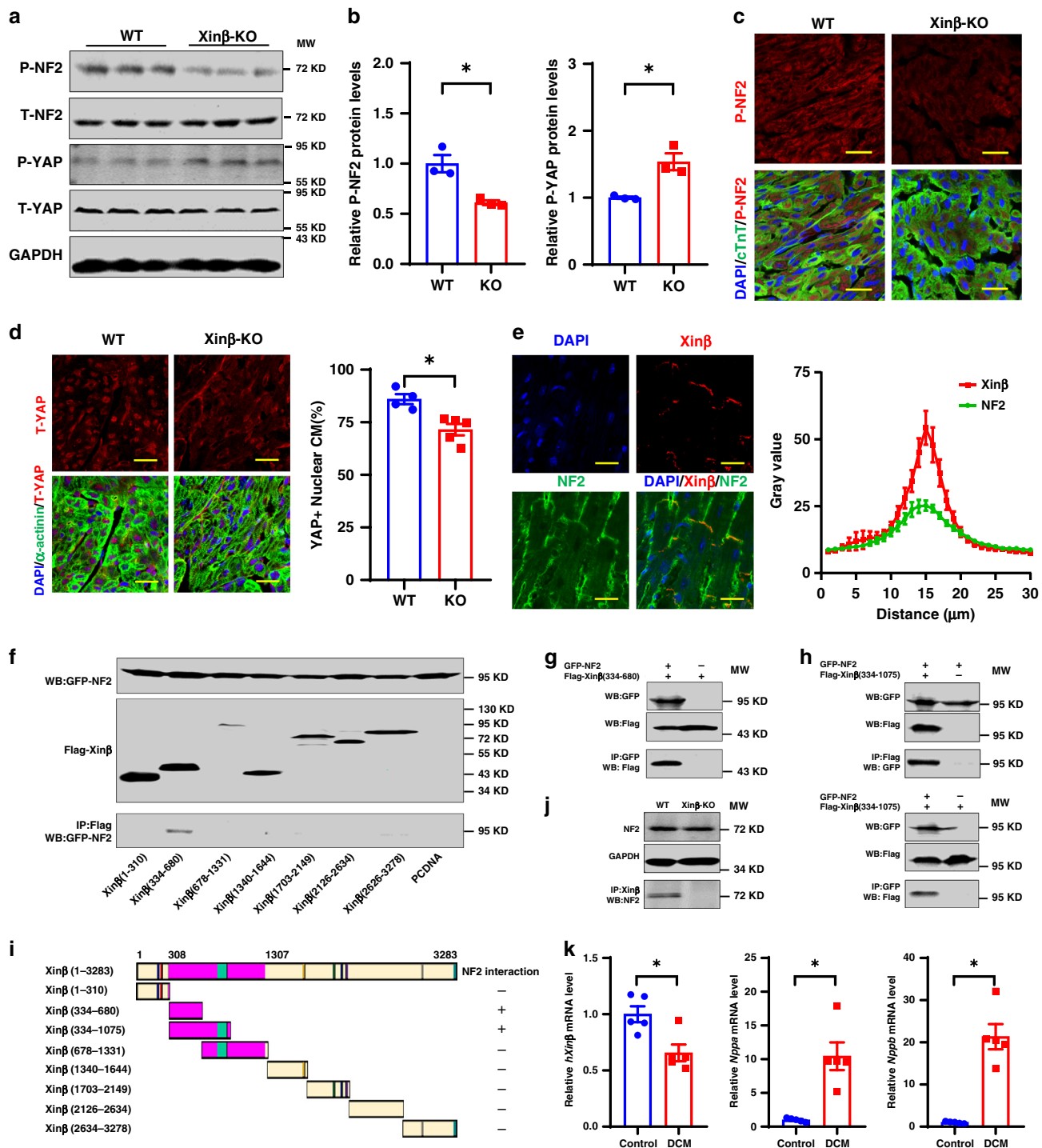

**Fig. 4 Xinβ interacts with NF2 to modulate the Hippo-YAP signaling in the heart. a** Western blots of indicated proteins using heart samples from P7.5 *Xinβ*[KO] and control mice. **b** Quantification of p-NF2 and p-YAP protein levels in P7.5 *Xinβ*[KO] and control hearts. $N = 3$ biologically independent samples, *$P <$ 0.05. **c** Immunohistochemistry detecting p-NF2 in P7.5 *Xinβ*[KO] and control hearts. DAPI marks nuclei, cTnT labels cardiomyocytes. Scale bars = 20 μm. **d** Immunohistochemistry detecting total YAP (t-YAP) in P7.5 *Xinβ*[KO] and control hearts. DAPI marks nuclei, α-actinin labels cardiomyocytes. Scale bars = 20 μm. Quantification of nuclear located YAP-positive cardiomyocytes is shown in the right panel. $N = 5$ and 4 biologically independent samples, respectively, *$P < 0.05$. **e** Immunohistochemistry detecting NF2 and Xinβ expression in the intercalated discs of adult cardiomyocytes. Scale bars = 20 μm. Quantification is presented in the right panel. $N = 11$ biologically independent samples, Data are presented as mean values ± SEM. **f** Co-immunoprecipitation assays detecting interactions between GFP-tagged NF2 and Flag-tagged Xinβ protein fragments. 5% cell lysate was used as input to demonstrate the expression of tagged proteins. **g** Co-immunoprecipitation assay detecting the interaction between GFP-tagged NF2 and Flag-tagged Xinβ protein fragment (amino acids 334-680). Five percent cell lysate was used as input to demonstrate the expression of tagged proteins. **h** Co-immunoprecipitation assays detecting the interaction between GFP-tagged NF2 and Flag-tagged Xinβ protein fragment (amino acids 334-1075) using antibodies to Flag (top) and GFP (bottom). Five percent cell lysate was used as the input to demonstrate the expression of tagged proteins. **i** Summary of the Xinβ domains that interact with NF2. **j** Interaction of endogenous Xinβ with NF2 protein. **k** qRT-PCR of human *Xinβ* (*hXinβ*) and cardiomyopathy marker genes *NPPA* and *NPPB* in heart samples from dilated cardiomyopathy (DCM) and control patients. $N = 5$ biologically independent samples, *$P < 0.05$.

active YAP (AAV-*YAP*[S127A]). One of the reasons for the partial rescue is likely due to the need to rapidly express *YAP*[S127A] to fully overcome the cardiac defects due to loss of Xinβ. AAV-mediated gene expression increases over several days following vector delivery and the level required for functional rescue may not have been achieved in mice that were not completely rescued. Alternatively, active YAP may only partially mediate the function of Xinβ in postnatal mouse hearts, and other unidentified molecular pathways may be involved. Given the fact that rescued *Xinβ*[KO] mice display a spectrum of phenotypes, including normalization of cardiac function and gene expression, we believe that we could achieve a full rescue if YAP expression was initiated at an earlier time.

Extensive studies have recently established important roles of the Hippo-YAP signaling pathway in heart development, regeneration, and cardiomyopathy[15,18]. Using gain- and loss-of-function genetic models, it was shown that the Hippo-YAP signaling pathway components Mst1/2, Sav1, Lats1/2, NF2, and downstream transcriptional regulators YAP and Tead1 regulate cardiac gene expression, cardiomyocyte proliferation, and cardiac disease status[14,27,28]. Interestingly, a recent study showed that YAP can be recruited by dystrophin and sequestered to the cell membrane of cardiomyocytes, leading to inhibition of cardiomyocyte proliferation[16]. These findings indicate that it is possible that YAP is recruited by Xinβ to the ICD, resulting in inhibition of the Hippo-YAP signaling pathway. Precisely how Xinβ and the ICD modulate the Hippo-YAP signaling pathway remains to be fully understood.

The present studies demonstrate that Xinβ interacts with NF2 and that the loss of Xinβ results in a reduction of phosphorylated NF2, but not T-NF2 protein. These observations suggest that Xinβ potentially acts as a scaffold to recruit NF2 and mediate its phosphorylation near the surface of cardiomyocytes. As a consequence of Xinβ loss and decreased quantities of phosphorylated NF2, the protein levels of P-YAP increase in the *Xinβ*[KO] heart, leading to a decrease in nuclear YAP concentration in cardiomyocytes (Fig. 5). This mechanism appears to be different from that of a prior study where NF2 was reported to translocate to the nucleus, leading to repression of YAP-mediated gene expression in cardiomyocytes[23]. Therefore, the precise mechanism of Xinβ modulation of the Hippo-YAP pathway remains to be fully understood.

Our studies also associate the regulation of Xinβ with DCM in humans. While we found that hXinβ expression was reduced in DCM patient hearts, a recent study using a mouse model showed increases in expression of Xinα and Xinβ (Xirp1 and Xirp2)[29]. We speculate that this discrepancy may result from differences in disease stage because the sample collection time in the mouse study occurred 2 weeks after tamoxifen injection, which is

considered to be relatively early in terms of the pathological progression of DCM. Clearly, the role of these proteins in the progression of cardiac disease warrants further investigation. At the same time, our studies establish that Xinβ is a tissue-specific, membrane-associated protein that regulates the Hippo-YAP signaling pathway to control heart development and our findings imply that Xinβ may be a novel therapeutic target for the treatment of cardiovascular diseases.

## Methods

**Human samples.** Human samples were collected in accordance with the World Medical Association's Declaration of Helsinki and procedures were approved by the Institutional Ethics Committee of the National Institute of Cardiovascular Diseases, Bratislava, Slovakia. The research described here complied with all relevant ethical regulations for work with human participants, and patients provided written informed consent prior to tissue collection. Left ventricle (LV) tissues were taken from patients with terminal-stage heart failure indicated for heart transplantation[30]. In brief, the patient's heart was removed at the time of transplantation, and LV tissue was subsequently dissected and snap frozen in liquid nitrogen. We used LV samples from healthy hearts that were not implanted to serve as controls.

**Mouse strains.** The research described here complied with all relevant ethical regulations for animal testing and research. All animal strains, procedures, and ethical considerations were approved by the Institutional Animal Care and Use Committee at Boston Children's Hospital. Xinβ-knockout mice (*Xinβ*[KO]) were created by constructing a targeting vector to replace portions of exon 6, intron 6, and a portion of exon 7 with a LacZ-Neor cassette[4]. The linearized targeting vector was electroporated into R1 embryonic stem (ES) cells at the University of Iowa Gene Targeting Facility. After selection, G418-resistant ES clones were screened for the presence of the targeted locus and positive clones were expanded and then micro-injected into C57BL6 blastocysts to generate chimera. Heterozygous mice were back-crossed to C57BL6 for at least seven generations and maintained in C57BL6 background. Rosa[Cas9GFP/Cas9GFP] mice were acquired from the Jackson Laboratory (No.026175)[31]. Studies with the *Xinβ*[KO] strain used male mice and samples used for RNA-seq had an equal mixture of male and female samples in each group. The ages of the mice used in the studies are listed in the figure legends and text. All comparisons used littermates as controls.

**Echocardiography.** Echocardiographic measurements were performed on mice using a Vevo 2100 Imaging System (Visual Sonics) with a 40-MHz MicroScan transducer (model MS-50D). Mice were anesthetized with isoflurane (2.5% isoflurane for induction and 0.1% for maintenance). Heart rate and LV dimensions, including diastolic and systolic wall thicknesses, LV end-diastolic and end-systolic chamber dimensions were measured from 2-D short-axis under M-mode tracings at the level of the papillary muscle. LV mass and functional parameters such as percentage of fractional shortening (FS%) and ejection fraction (EF%) were calculated using the above primary measurements and accompanying software[32].

**Generation and administration of AAV9.** *GFP* and 3Flag-h*YAP*[S127A] were separately cloned into ITR-containing AAV plasmid (Penn Vector Core P1967) harboring the chicken cardiac TNT promoter, to get pAAV.*cTnT*:3*Flag*-h*YAP*[S127A] and pAAV.*cTnT*:*GFP*. AAV9 was packaged in HEK293T cells with AAV9:Rep-Cap and pHelper (pAd deltaF6, Penn Vector Core), then purified and concentrated by gradient centrifugation[17,33]. AAV9 titer was determined by quantitative PCR. AAV9 virus ($1 \times 10^{10}$ viral genomes [vg]/g) was injected into postnatal day 1

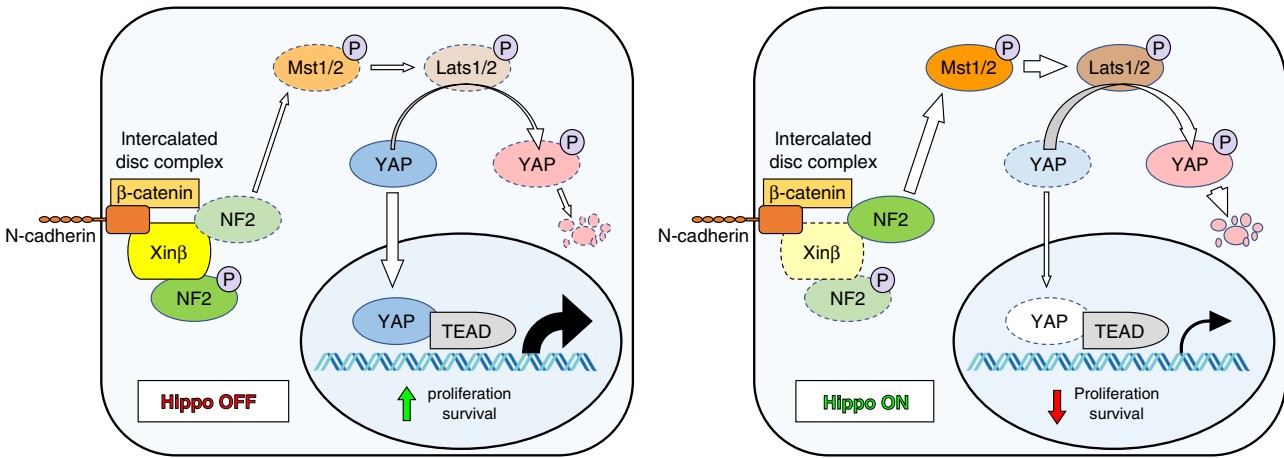

**Fig. 5 Model of Xinβ regulation of the Hippo-YAP signaling pathway in cardiomyocytes.** Xinβ negatively modulates Hippo-YAP signaling by recruiting phosphorylated NF2 to the sarcolemmal membrane of cardiomyocytes, leading to the reduction of total NF2 protein, repression of Hippo signaling, and the activation of the downstream mediator YAP (left panel). Loss of Xinβ results in activation of Hippo signaling and a reduction of functional YAP levels, which reduces expression of downstream targets.

Xinβ[KO] pups or their control littermates with subcutaneous injection. Pup hearts were harvested at the age of postnatal day 7.5 for Xinβ[KO] mice. For CASAAV-mediated Xinβ gene depletion, we designed two gRNAs using the GPP Web Portal (Broad Institute). The gRNA sequences were synthesized as single-stranded oligonucleotides, annealed, and inserted into AAVU6gRNA-U6gRNA-cTNT-Cre plasmids at SapI and NheI sites. AAV9 was packaged in HEK293T cells and then purified and concentrated by gradient centrifugation. $5 \times 10^{10}$-vg/g AAV was injected into Rosa[Cas9GFP/Cas9GFP] mice P1 pups subcutaneously.

**Histology.** Mouse heart tissues were dissected, rinsed with PBS, and fixed in 4% paraformaldehyde (pH 7.4) overnight. After dehydration through an ethanol series, samples were embedded in paraffin wax according to standard laboratory procedures.

Sections of 5 μm were stained with Hematoxylin and Eosin or further fixed with pre-warmed Bouin's solution at 55 °C for 1 h and stained with Fast Green and Sirius Red as previously described[34]. The stained sections were used for routine histological examination by light microscopy and quantified with ImageJ software.

**Immunofluorescence.** Mouse hearts were dissected, collected, and fixed in 4% PFA at 4 °C for 4 h. After washing in PBS, samples were treated in 30% sucrose for 12 h each and embedded in OCT. About 8-μm cryostat sections were collected on positively charged slides. Sections were washed in PBS, blocked in 5% serum/PBS, subjected to immunostaining with anti-mouse Xinβ (U1040) primary antibody overnight at 4 °C, and then washed three times with PBS buffer before adding IgG secondary antibodies conjugated to Alexa Fluor 488, 594, or 647 (ThermoFisher Scientific). Primary and secondary antibodies are listed in Supplementary Data 1 (Table 1). EdU was detected with the Click-iT EdU imaging kit (Invitrogen, C10339) as previously described[35]. TUNEL staining was performed using the ApopTag Peroxidase In Situ Apoptosis Detection Kit (Millipore, S7110)[19]. Fluorescently stained cells were counterstained with DAPI. Imaging was performed on a FV3000 confocal (Olympus) or BZ-X710 automated epi-fluorescent (Keyence) microscope.

**Cardiomyocyte culture.** Neonatal rat ventricular myocytes (NRVMs) were isolated by enzymatic disassociation of 1-day-old rat hearts using the Neonatal Cardiomyocyte Isolation System (Cellutron Life Technology)[36]. Cardiomyocytes were pre-plated for 2 h to remove fibroblasts. Cells were then plated on 1% gelatin-coated plates in medium containing 10% horse serum and 5% fetal calf serum. For the treatment of siRNA, fifty 50 nM of siRNA targeting XIRP2 transcript (siXinβ) and control siRNA (Dharmacon) were transfected into cardiomyocyte by using Lipofectamine RNAiMAX transfection reagent. Six hours later, medium with transfection reagent was removed. Twenty-four hours after plating, cells were fed serum-free medium and infected with adenovirus (Ad-GFP for control and AdYAP1) at 25 MOI for 24 h.

**Isolation of cardiomyocytes from adult mice.** Adult mouse cardiomyocytes were isolated by perfusing and digesting hearts with collagenase II (Worthington Biochemical Corp.)[37]. Dissociated cells (i.e., containing cardiomyocytes and non-myocytes) were allowed to sediment by gravity. The bottom layer contained adult cardiomyocytes, which were collected and fixed with 4% PFA for immuno-fluorescence staining.

**Reverse transcription and quantitative PCR analysis.** Total RNAs were isolated using TRIzol reagent (ThermoFisher Scientific) from cells and tissue samples. For quantitative RT-PCR, 2-μg RNA samples were reverse-transcribed to cDNA by using random hexamers and MMLV reverse transcriptase (ThermoFisher Scientific) in a 20-μL reaction. In each analysis, 0.1-μL cDNA pool was used for quantitative PCR. Real-time PCR was performed using an ABI 7500 thermocycler with the Power SYBR Green PCR Kit (ThermoFisher, 4368702). The relative expression of interested genes is normalized to the expression of 18S rRNA or β-actin. Quantitative RT-PCR primers are listed in Supplementary Data 1 (Table 2).

**Western blot analysis.** Protein lysates were prepared from heart tissues in RIPA buffer supplemented with proteinase and phosphatase inhibitors (ThermoFisher). Lysate samples were cleared by 10,000 g centrifugation for 10 min. Samples were subsequently analyzed by SDS/PAGE and transferred to PVDF membranes that were incubated with Odyssey Blocking Buffer (LI-COR) and primary antibody overnight at 4 °C and then washed three times with PBS buffer before adding IgG secondary antibody. GAPDH or β-tubulin were used as a loading control. Specific protein bands were visualized with the Odyssey imaging system. Antibodies are listed in Supplementary Data 1 (Table 1) and full scan blots are available in the Source Data.

**Co-IP assays.** HEK293T cells were transiently transfected with plasmids using Lipofectamine 3000 (Life Technologies). Cells were harvested 48 h after transfection in lysis buffer composed of PBS containing 0.5% Triton X-100, 1-mM EDTA, 1-mM PMSF, and Complete Protease Inhibitor Cocktail Tablets (Sigma-Aldrich). After a brief sonication and removal of debris by centrifugation, proteins were precipitated with Anti-Flag, Anti-GFP antibodies, and protein A/G beads and analyzed by western blotting.

**RNA sequencing and data analysis.** Total RNA from the cardiac ventricles of 7.5-day-old mice was isolated using TRIzol (ThermoFisher Scientific) and strand-specific poly A-selected libraries were generated with the TruSeq RNA Library Preparation Kit (Illumina). For the Xinβ-KO and WT experiment, the obtained libraries were combined in equimolar amounts and loaded on a single Illumina flow cell lane, followed by single-ended sequencing (SE50bp, TUCF genomic facility). For the AAV9-GFP or AAV9-YAP-injected Xinβ-KO and WT experiment, the obtained libraries were combined in equimolar amounts and loaded on a single Illumina flow cell lane, follow by paired-ended sequencing (PE150bp, GENEWIZ). FASTQ files were extracted and the TruSeq sequencing adapters and low-quality reads were removed from FASTQ files with Cutadapt v.2.3. The cleaned FASTQ files were quality checked using FastQC (Babraham Bioinformatics), then aligned to the mouse genome (Esembl GRCm38 genome obtained from GENCODE) using HISAT2 (v.2.1.0)[38]. Subsequently, transcript assembly was performed using StringTie (v.1.3.4)[39] with the annotated transcriptome as a reference. The assembled transcriptomes were quantified using prepDE.py script provided by the StringTie developer to generate gene matrix files. EdgeR (v.3.26.1)[40,41] was used to compute counts per million (CPM) as a normalized measurement for gene expression.

Differentially expressed genes were tested using the Fisher's exact test, and multiplicity correction is performed with the Benjamini–Hochberg method on the P values, to control the false discovery rate (FDR). The exact P value and FDR q value for the differential gene expression analysis can be found in the Source

Data. Differentially regulated genes with FDR value <0.05 were considered significant. We defined expressed genes as those that have expression in at least half of all samples. Expressed genes were subjected to PCA. Principal component 1 and 2 were plotted in 2-D coordinates. Deposited Gene Expression Omnibus (GEO) RNA-seq genomic data are publically available (accession number GSE149647) and contained in the Source Data.

**Gene set enrichment analysis (GSEA)**. GSEA was performed on expressed genes according to the software manual[42,43]. Gene sets with a nominal p value of <0.05 and an FDR of <0.25 were considered significant. Exact nominal P value and FDR q value for the GSEA analysis can be found in the Source Data. All expressed genes were Log2 transformed, centered, and unsupervised hierarchical clustering was performed using the k-mean clustering method with Cluster 3.0 software[44]. Java Treeview (v.3.0) was used to visualize the clustered heatmaps.

**Microarray data analysis**. Active YAP (i.e., aYAP also referred to as YAP^{S127A}) overexpression in the NRVM microarray data[45] was obtained from GEO accession number GSE57719. Raw CEL files were analyzed using Transcriptome Analysis Console (Applied Biosystems) for differential gene expression. Differentially regulated gene with Benjamini–Hochberg FDR < 0.05 were considered significant.

**Statistics**. Values are reported as means ± SEM unless indicated otherwise. One-way analysis of variance analysis followed by Dunnett's post hoc testing was used to evaluate the statistical significance for multiple-group comparisons. In addition, the two-tailed nonparametric Student's T-test was used for two-group comparisons. Values of P < 0.05 were considered statistically significant. Data are presented as mean values ± SEM. Exact P values can be found in the Source Data.

**Reporting summary**. Further information on research design is available in the Nature Research Reporting Summary linked to this article.

## Data availability

All relevant data related to this manuscript are available upon reasonable request from the authors. The RNA-seq data described in this study has been deposited to the Gene Expression Omnibus (GSE) with the accession number GSE149647. Source Data are provided with this paper. Original un-cropped western blots and raw numbers used for the statistics presented in the Figures and Supplementary Information are contained in the Source Data. A list of the antibodies and primer sequences used in this study are provided in Supplementary Data 1. Source data are provided with this paper.

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

## Acknowledgements
We thank members of the Wang Laboratory for their advice and support. Work in the Wang Lab is supported by the NIH (HL149401, HL138757, and HL125925). Z.-P. Huang was supported by NIH T32HL007572.

## Author contributions
H.G. and D-.Z.W. conceived the project, designed, and analyzed the experiments. H.G., H.Y.S., and Z-.P.H. performed molecular biology analyses. J.K. contributed to human sample acquisition and cDNA library preparation. Y.W.L. performed bioinformatics. J.L. and J.J.C.L. generated Xinβ$^{KO}$ mice and antibodies. H.G., K.L., and Q.M. performed animal studies. H.G., Y.W., and Y.W.L. contributed to the echocardiographic data acquisition and analysis. H.G. and Z.L. contributed to adeno-associated virus preparation and administration. H.G. and X.H. contributed to the histological and immuno-fluorescent data acquisition and analysis. W.T.P. supervised the YAP studies. H.G., Y.W.L., D.B.C., and D-.Z.W. wrote the manuscript.

## Competing interests
The authors declare no competing interests.
