## [Peer Review File · Nature Communications]

REVIEWER COMMENTS

Reviewer #1 (Remarks to the Author):

The authors study the role of Xinb, a component of ICDs. They find that deletion of Xinb results in more Hippo pathway activity and this occurs through a direct interaction between Xinb and NF2 although the mechanism underlying this remains somewhat obscure.

Major comments:

- 1) RNA seq – confirm the cell type or is the data derived from whole heart?
- 2) Is this a CM specific knock out? Please be clear about this
- 3) The majority of the data involves manipulations at post natal stages. Can the authors confirm the status of the ICD at these early stages postnatally and show some data. My impression is that the ICD is not completely formed until later stages in the more mature heart. This is an important point because much of the data may not actually be due to the function of Xinb in the ICD.
- 4) I find the presentation of data in figure 1 I and j somewhat confusing – please reevaluate the data presentation for this section
- 5) The final mechanism remains unclear to me. Xinb interacts with NF2 and this results in less inactive P-NF2 in the Xinb KO and thus more Hippo pathway activity with more p-Yap. How does this work?

Reviewer #2 (Remarks to the Author):

This manuscript investigates the role of the intercalated disc protein Xin beta in Hippo-Yap signalling in the heart. The authors show that one of the changes in Xin beta KO hearts in GSEA is the down regulation of YAP pathway genes. Also, there seems to be the opposite response in gene expression in the Xin beta KO scenario compared to a YAP-overexpressing one. Since Hippo-YAP signaling is a major control factor in the regulation of proliferation in cardiomyocytes, the authors analysed this in their KO mice and found smaller hearts with fewer EdU positive cardiomyocytes. Postnatal KO of Xin beta using an AAV9 mediated CRISPR/Cas9 strategy with the appropriate gRNAs led to a 80% reduction of Xin beta expression, but in contrast to the conventional KO mice, the mice did not die before weaning and Xin negative cardiomyocytes looked normal. This suggests that the major input of the Xin-Hippo-YAP signaling axis is for hyperplastic growth of the heart, but that its reduced activity can be tolerated in the adult heart. Overexpression of a constitutively active version of YAP (S127A) via AAV9 at postnatal day 1 rescued 40% of the Xin beta KO mice, mainly by influencing cardiomyocyte proliferation activity. They then investigated, who provides the link between the intercalated disc and the Hippo pathway and identified an interaction between Xin beta and NF2.

This manuscript is extremely interesting since it provides another piece of evidence for the relevance of Hippo-YAP signalling in regulating the proliferation activity of cardiomyocytes and links this to a bona fide intercalated disc protein such as Xin beta. However, there are a few misleading or unclear statements that should be corrected in a revised manuscript.

My main problem is the claim for a DIRECT interaction between Xin beta and NF2 based on co-immunoprecipitation experiments, because this claim cannot be justified. Both Xin beta as well as NF2 were shown to bind to a plethora of other proteins (e.g. Xin bds beta-catenin but also nebulin; Eulitz et al., 2013; interactome for NF2 in other cell types analysed by Hennigan et al., 2019). Therefore their results indicate that the two proteins are part of a complex, but direct interaction remains to be shown. Hence this must be corrected throughout the manuscript.

Also the name of the AAV9-YAP should be changed throughout manuscript and figures to clearly indicate that this is a constitutively active version (either call it S127A or GOF) and not wild type YAP.

The authors find reduced transcripts for Xin beta in human DCM patients (Figure 4i). However just recently Zhou et al. identified a massive up-regulation of Xirp1 and Xirp2 in their DCM model, suggesting this might be a hallmark of DCM. This should be discussed (Zhou et al., 2020; Circulation).

In their case the mice seem to tolerate the expression of a constitutively active YAP, while for Monroe et al. 2019 the mice died within a week of over-expression. Slightly different experimental approaches, but again, this should be discussed.

Clarify that the presence of Edu and pH3 is not conclusive evidence for cell proliferation but could also indicate nuclear division.

In Figure 2g there are two extra nuclei stuck to the Xin beta cell; this might be confusing; select a better image.

Figure 3c: The KO mouse is smaller but do the authors have any explanation why it also has less fur?

RESPONSE TO THE REVIEWERS' COMMENTS

Reviewer #1

The authors study the role of $Xin\beta$, a component of ICDs. They find that deletion of $Xin\beta$ results in more Hippo pathway activity and this occurs through a direct interaction between $Xin\beta$ and NF2 although the mechanism underlying this remains somewhat obscure.

Response: We appreciate the Reviewer's input and we have revised our manuscript to address the Reviewer's comments. Specifically, we have modified the text to clarify the mechanism of Hippo-YAP activation through $Xin\beta$ interaction with NF2 and the ICD. Below, we have provided point-by-point responses to each comment.

Major comments:

1) RNA seq – confirm the cell type or is the data derived from whole heart?

Response: The RNA-seq data we generated from the KO studies were derived from the ventricle tissues of P7.5 heart.

2) Is this a CM specific knock out? Please be clear about this

Response: For the neonatal study and rescue experiment, we used the germline $Xin\beta$ knockout line that we generated previously. For the adult study, we use the CasAAV system to generate the postnatal knockout in a cardiac-specific manner, as we have previously reported (Guo, Y., et al., *Analysis of Cardiac Myocyte Maturation Using CASAAV, a Platform for Rapid Dissection of Cardiac Myocyte Gene Function In Vivo*. *Circ Res*, 2017 **120**(12): p. 1874-1888; VanDusen, N.J., et al., *CASAAV: A CRISPR-Based Platform for Rapid Dissection of Gene Function In Vivo*. *Curr Protoc Mol Biol*, 2017. **120**: p. 31.11.1-31.11.14). We have clarified this in the revision.

3) The majority of the data involves manipulations at post-natal stages. Can the authors confirm the status of the ICD at these early stages postnatally and show some data. My impression is that the ICD is not completely formed until later stages in the more mature heart. This is an important point because much of the data may not actually be due to the function of $Xin\beta$ in the ICD.

Response: Numerous prior works have described the development and maturation process of the ICD and, indeed, it is generally believed that no mature ICD structure is formed before a neonatal age of 2 weeks in rodent hearts (Alexia Vite & Glenn L. Radice *N-Cadherin/Catenin Complex as a Master Regulator of Intercalated Disc Function*. *Cell Communication & Adhesion*, 2014 **21**(3): p. 169-179; Vermij SH, Abriel H, van Veen TA. *Refining the molecular organization of the cardiac intercalated disc*. *Cardiovasc Res*. 2017 **113**(3): p. 259-275). We have previously

reported the expression levels and cellular location of the $Xin\alpha$ and $Xin\beta$ proteins in developmental, neonatal, and adult hearts (Wang Q, Lin JL, Chan SY, Lin JJ. 2013). The Xin repeat-containing protein, $mXin\beta$, initiates the maturation of the intercalated discs during postnatal heart development (Dev Biol. 2013 Feb 15;374(2):264-80). We found that the ICD failed to mature in neonatal cardiomyocytes (between p7.5 to p24.5) of $mXin\beta$ deficient hearts, evidenced by the failure of restriction of ICD proteins (N-cadherins) to termini of cardiomyocytes. We have confirmed these observations (shown below). We found that connexin 43 and desmoplakin proteins are ‘mislocated’ to the lateral sides of cardiomyocytes in P14.5 $mXin\beta$ -KO hearts, whereas they are enriched to the termini of cardiomyocytes in control wild type mouse hearts. In contrast, the loss of $mXin\beta$ did not seem to significantly alter the expression and distribution of these protein at P7.5, before the maturation of ICD. We don’t feel that these data should be added into our manuscript since a similar observation has already reported by us previously. Nevertheless, we have added text to the Discussion in the revision to clarify this important issue.

4) I find the presentation of data in figure 1 I and j somewhat confusing – please reevaluate the data presentation for this section

Response: We have removed figure 1 I and simplified figure 1 J (new figure 1 I) to make the data presentation more direct and clearer.

5) The final mechanism remains unclear to me. Xinb interacts with NF2 and this results in less inactive P-NF2 in the Xinb KO and thus more Hippo pathway activity with more p-Yap. How does this work?

Response: Thank you for pointing this out and we agree. As presented in our studies and described in our working model (Figure 5), we found that loss of Xin β dramatically altered the Hippo-YAP pathway in a manner that an active form of YAP is sufficient to rescue the Xin β loss of function phenotype in mice. Our findings suggest that Xin β may act as a scaffold to recruit NF2 and mediate its phosphorylation on the surface of cardiomyocytes. However, exactly how Xin β modulates the Hippo-YAP pathway remains fully understood and that will be one of the focuses of future investigation. We have added additional discussion in the revision.

Reviewer #2

This manuscript investigates the role of the intercalated disc protein Xin beta in Hippo-Yap signalling in the heart. The authors show that one of the changes in Xin beta KO hearts in GSEA is the down regulation of YAP pathway genes. Also, there seems to be the opposite response in gene expression in the Xin beta KO scenario compared to a YAP-overexpressing one. Since Hippo-YAP signaling is a major control factor in the regulation of proliferation in cardiomyocytes, the authors analysed this in their KO mice and found smaller hearts with fewer EdU positive cardiomyocytes. Postnatal KO of Xin beta using an AAV9 mediated CRISPR/Cas9 strategy with the appropriate gRNAs led to a 80% reduction of Xin beta expression, but in contrast to the conventional KO mice, the mice did not die before weaning and Xin negative cardiomyocytes looked normal. This suggests that the major input of the Xin-Hippo-YAP signaling axis is for hyperplastic growth of the heart, but that its reduced activity can be tolerated in the adult heart. Overexpression of a constitutively active version of YAP (S127A) via AAV9 at postnatal day 1 rescued 40% of the Xin beta KO mice, mainly by influencing cardiomyocyte proliferation activity. They then investigated, who provides the link between the intercalated disc and the Hippo pathway and identified an interaction between Xin beta and NF2.

This manuscript is extremely interesting since it provides another piece of evidence for the relevance of Hippo-YAP signalling in regulating the proliferation activity of cardiomyocytes and

links this to a bona fide intercalated disc protein such as Xin beta. However, there are a few misleading or unclear statements that should be corrected in a revised manuscript.

Response: We appreciate the Reviewer's enthusiasm for our work and we have revised the manuscript to more accurately reflect our findings. Below, we have provided point-by-point responses to each comment.

My main problem is the claim for a DIRECT interaction between Xin beta and NF2 based on co-immunoprecipitation experiments, because this claim cannot be justified. Both Xin beta as well as NF2 were shown to bind to a plethora of other proteins (e.g. Xin bds beta-catenin but also nebullette; Eulitz et al., 2013; interactome for NF2 in other cell types analysed by Hennigan et al., 2019). Therefore their results indicate that the two proteins are part of a complex, but direct interaction remains to be shown. Hence this must be corrected throughout the manuscript.

Response: We have modified the text to more accurately reflect the nature of the Xin β interaction.

Also the name of the AAV9-YAP should be changed throughout manuscript and figures to clearly indicate that this is a constitutively active version (either call it S127A or GOF) and not wild type YAP.

Response: We have adjusted the nomenclature in the revised manuscript to clarify that we have employed a constitutively active AAV construct (*i.e.* 'aYAP' indicates the constitutively active YAP). We have also standardized the name of the knock-out mouse strain.

The authors find reduced transcripts for Xin beta in human DCM patients (Figure 4i). However just recently Zhou et al. identified a massive up-regulation of Xirp1 and Xirp2 in their DCM model, suggesting this might be a hallmark of DCM. This should be discussed (Zhou et al., 2020; Circulation).

Response: We speculate that the up-regulation of Xirp1 and Xirp2 in the mouse model in Zhou et al maybe due to the stage of disease progression. We noticed that in Zhou et al. the sample collection time was from 2-weeks post tamoxifen injection, which is relatively early in terms progression of the pathology. To check whether our qPCR result on human samples is consistent with independent cohorts, we re-analyzed an independent set of data from a pre-print manuscript (Spurrell *et al.*, 2019 BioRxiv), where they had profiled large number of human idiopathic dilated cardiomyopathy (iDCM) and healthy individuals (Figure below). We found that in this cohort of samples, Xirp2 expression was downregulated in iDCM samples, consistent with our qPCR result.

Citation: Spurrell CH, Barozzi I, Mannion BJ, et al. Genome-Wide Fetalization of Enhancer Architecture in Heart Disease. *bioRxiv*; 2019. DOI: 10.1101/591362.

In their case the mice seem to tolerate the expression of a constitutively active YAP, while for Monroe et al. 2019 the mice died within a week of over-expression. Slightly different experimental approaches, but again, this should be discussed.

Response: We appreciate the Reviewer's suggestion. In Monroe *et al.*, 2019, the version of active YAP being used have all five Ser residues mutated to Ala (YAP S5A), which is a much more potent form of active YAP. However, we used the S127A YAP, which we previously reported to not affect viability of the mice (Lin, Z., et al., *Cardiac-specific YAP activation improves cardiac function and survival in an experimental murine MI model*. *Circ Res*, 2014. **115**(3): p. 354-63). These are two very different forms of constitutive YAP.

Clarify that the presence of Edu and pH3 is not conclusive evidence for cell proliferation but could also indicate nuclear division.

Response: We have modified the text to include the possibility that our data may indicate nuclear division and/or cell proliferation.

In Figure 2g there are two extra nuclei stuck to the Xin beta cell; this might be confusing; select a better image.

Response: We agree with the Reviewer's observation and we have replaced Figure 2G.

Figure 3c: The KO mouse is smaller but do the authors have any explanation why it also has less fur?

Response: We believe this difference is a reflection of a developmental delay. As shown below, we noticed that at a later stage, AAV-YAP-injected $Xin\beta$ -KO mouse is indistinguishable from WT control in size.

REVIEWERS' COMMENTS:

Reviewer #1 (Remarks to the Author):

this is an improved manuscript. mechanisms for regulation of Hippo-Yap at the intercalated disk are very poorly understood and this paper adds important new insights. The paper will be of interest to those in cardiovascular area as well as the Hippo pathway field

Reviewer #2 (Remarks to the Author):

My questions have all been addressed in a satisfactory fashion by the authors in the revised manuscript and by the additional data in the rebuttal letter.